# Differential Radiosensitizing Effect of 50 nm Gold Nanoparticles in Two Cancer Cell Lines

**DOI:** 10.3390/biology11081193

**Published:** 2022-08-09

**Authors:** Miguel Ángel Pérez-Amor, Leonardo Barrios, Gemma Armengol, Joan Francesc Barquinero

**Affiliations:** 1Unit of Biological Anthropology, Department of Animal Biology, Plant Biology and Ecology, Universitat Autònoma de Barcelona, 08193 Bellaterra, Catalonia, Spain; 2Unit of Cell Biology, Department of Cell Biology, Physiology and Immunology, Universitat Autònoma de Barcelona, 08193 Bellaterra, Catalonia, Spain

**Keywords:** gold nanoparticles, functionalization, internalization, radiation therapy, γ-H2AX foci, radiosensitization

## Abstract

**Simple Summary:**

Nanoparticle treatment on tumor cells is proposed for its potential radiosensitizing properties, increasing the radiation effect on tumor cells and reducing the adverse effects on healthy tissues. The present study evaluates, on two cell lines derived from colon and breast adenocarcinomas, the impact of irradiation in the presence of specifically targeted gold nanoparticles. Cells were irradiated in the absence and in the presence of non-functionalized or specifically functionalized gold nanoparticles. The results pointed out that actively targeting gold nanoparticles has a clear radiosensitizing effect in both cell lines.

**Abstract:**

Radiation therapy is widely used as an anti-neoplastic treatment despite the adverse effects it can cause in non-tumoral tissues. Radiosensitizing agents, which can increase the effect of radiation in tumor cells, such as gold nanoparticles (GNPs), have been described. To evaluate the radiosensitizing effect of 50 nm GNPs, we carried out a series of studies in two neoplastic cell lines, Caco2 (colon adenocarcinoma) and SKBR3 (breast adenocarcinoma), qualitatively evaluating the internalization of the particles, determining with immunofluorescence the number of γ-H2AX foci after irradiation with ionizing radiation (3 Gy) and evaluating the viability rate of both cell lines after treatment by means of an MTT assay. Nanoparticle internalization varied between cell lines, though they both showed higher internalization degrees for functionalized GNPs. The γ-H2AX foci counts for the different times analyzed showed remarkable differences between cell lines, although they were always significantly higher for functionalized GNPs in both lines. Regarding cell viability, in most cases a statistically significant decreasing tendency was observed when treated with GNPs, especially those that were functionalized. Our results led us to conclude that, while 50 nm GNPs induce a clear radiosensitizing effect, it is highly difficult to describe the magnitude of this effect as universal because of the heterogeneity found between cell lines.

## 1. Introduction

Radiation therapy is one of the most common treatments when fighting neoplasms and almost 60% of patients receiving anti-neoplasic treatment undergo radiation therapy [1], be it alone or concomitant with another type of treatment, i.e., chemotherapy or surgery. Despite its effectiveness in cancer treatment, exposure to radiation therapy can cause several adverse side effects, such as dermatitis, fatigue, depression and radiation recall [2].

With the aim of either diminishing the appearance of undesired side effects or increasing the radiation effect in tumoral cells, radioprotective and radiosensitizing agents can be used. A radioprotective agent reduces the damage produced in healthy cells by limiting the action on DNA of reactive radicals such as reactive oxygen species (ROS), mainly generated by the interaction of ionizing radiations and water [3]. A pioneering study showed that the fraction of the DNA damage that can be protected from radiation by the use of radioprotectors accounts for about 70% [4]. Radioprotective agents such as amifostine are already being used clinically [5], while others such as resveratrol and CBLB502 are being tested with promising results [6]. On the contrary, one of the main mechanisms of action of radiosensitizing agents is to enhance the level of radiation-induced ROS to cause damage to the DNA of tumoral cells. As radiosensitizing agents, various agents such as hyperbaric oxygen [7], nicotinamide and carbogen [8] are widely described and applied. A third type of agent exists, radiation mitigators [8], but their action is to reduce the impact of the adverse effects after an irradiation.

An important limitation of some radiosensitizing treatments can be attributed to the unspecific distribution of the radiosensitizer throughout the body. One strategy for overcoming this limitation is the use of micro- or nanoparticles as radiosensitizers because they can naturally accumulate in tumor tissues due to the enhanced permeability and retention effect resulting from the imperfect endothelium of blood vessels formed during tumoral angiogenesis [9]. To increase the particles’ accumulation in specific desired tissues, active targeting strategies have been developed, such as the functionalization of the particle surface to be recognized by cell membrane receptors [10] even in fluidic conditions [11].

The effect of nanoparticles as radiosensitizers has been extensively reviewed [12,13,14,15,16], and their importance as therapeutic agents has been steadily rising with time. It has been described that high atomic number (Z) materials absorb more energy per unit mass than water when irradiated with X-rays [16]. In the case of gold (Z = 79), it can be 100 times more effective at absorbing photon energy than water. This local absorption triggers the emission of low-energy electrons from gold with a potential increase in DNA damage. The increase in the effect of a dose when it is delivered in the presence of gold nanoparticles (GNPs) is called “sensitization enhancement ratio” (SER). So, if GNPs can accumulate in specific tissues, it opens the door to a differential enhancement in tumoral tissues [13], allowing for lower radiation doses to have the same effect as higher ones [17,18]. However, it has been observed that GNPs induce higher SER than could be expected according only to their physical conditions [16,19,20]. This means that several factors influence the radiosensitization effect of GNPs. In relation to energy deposition, low-energy X-rays produce a higher enhancement than high-energy X-rays [21,22] and, the higher the diameter of GNPs, the lower the deposition of energy and the emission of low-energy electrons [23]. On the other hand, the radiosensitizing effect of GNPs is, partially, triggered by an increase in the formation of reactive oxygen species (ROS) when compared with cells irradiated in the absence of GNPs. ROS can react with DNA, inducing double strand breaks and affecting the cell viability. Moreover, without irradiation GNPs also increase the oxidative stress of the cells by interfering in the activity of some antioxidant enzymes [24,25,26,27]. Because ROS have a limited lifespan, it seems reasonable that a greater radiosensitizing effect is expected if GNPs are inside the cell. A higher cell uptake of 50 nm diameter GNPs when compared to 14 nm or 74 nm GNPs has been described [22]. Considering all this information, to take advantage of the radiosensitizing effect of GNPs in radiotherapy, we think that an effective targeting system could be of great help. The aim of the present study was to evaluate the radiosensitizing effect of 50 nm spherical GNPs specifically functionalized to interact with cell membrane receptors (EpCAM and transferrin receptor) of two different cell lines, Caco2 (colon adenocarcinoma) and SKBR3 (breast adenocarcinoma), respectively. Caco2 cells have been described as highly radioresistant due to a low 53BP1 expression [28,29,30]. SKBR3 cells showed a survival fraction and a mean inactivation dose, after irradiation, close to the mean of eight breast cancer cell lines [31], but other studies described a lower intrinsic sensitivity to radiation due to the overexpression of HER2 [32,33]. Radiosensitizing effect was evaluated by means of DNA damage by analyzing radiation-induced γ-H2AX foci and cell viability. Nanoparticle internalization was also assessed using confocal microscopy.

## 2. Materials and Methods

### 2.1. Cell Lines 

In the present study, two different cell lines were used: Caco2 derived from colon adenocarcinoma cells (ATCC, Manassas, VA, USA) and SKBR3 derived from breast adenocarcinoma cells (ATCC). Both cell lines were cultured with Eagle’s modified enriched medium (EMEM, ATCC) and McCoy’s 5A medium (ATCC), respectively, at 37 °C and in 5% CO_2_ conditions. Doubling times were determined by counting stained cells with trypan blue (Sigma-Aldrich, Madrid, Spain) and visualizing by optical microscopy with the help of a hemocytometer (Sigma-Aldrich). Doubling times were 68 h for Caco2 cells and 63 h for SKBR3 cells.

### 2.2. Receptor Expression

To ensure that both cell lines behaved as expected and expressed the specific receptors (EpCam and TfR), an immunofluorescence staining was carried out. The primary antibodies used were Rabbit Anti-EpCam (cat. ab71916, Abcam, Cambridge, UK) and Mouse Anti-TfR (cat. ab9179, Abcam), and the secondary antibodies were Goat Anti-Rabbit IgG H&L Alexa Fluor^®^ 488 (cat. ab150077, Abcam) and Goat Anti-Mouse IgG H&L Alexa Fluor^®^ 488 (cat. ab150113, Abcam). First, cells were centrifuged and rinsed with 1 × PBS, then 35.000 cells were centrifuged using a Cytospin centrifuge (Cytospin 3, Shandon, Thermo Scientific, Madrid, Spain) and fixed to a polysine slide by applying 2% paraformaldehyde onto the sample. After 10 min, the slides were rinsed in 1 × PBS, and then fixed cells were permeabilized using 1 × PBS/0.5% Triton-100 at 4 °C. After 15 min, slides were again rinsed with 1 × PBS before applying a blocking solution (1 × PBS/0.1% Tween20/2% FCS) for 30 min. Then, cells were incubated with the primary antibody overnight in a humid chamber at 4 °C. Once incubation was complete, the slides were washed thrice for 5 min in a 1 × PBS/0.1% Tween20 solution to eliminate a possible primary antibody excess, then the secondary antibody was applied, and the slides were incubated for 1 h in a humid chamber at room temperature. After that, the slides were washed in 1 × PBS/0.1% Tween20 solution as previously described. Finally, 5 μL of Vectashield Mounting Medium (Vector Laboratories, Barcelona, Spain) with 4,6-diamidino-2-phenylindole (DAPI) as counterstain at a 1.5 μg·mL^−1^ concentration was applied.

### 2.3. Cytogenetic Characterization

To know the genomic stability of the cell lines used in the present study, first we analyzed the karyotypes of one hundred cells for each cell line. Fluorescence in situ hybridization techniques using pancentromeric and pantelomeric PNA probes were used to establish the modal karyotypes and to check for the presence of unstable chromosome aberrations. For this purpose, exponentially growing cell cultures were treated with colcemid at a final concentration of 0.3 μg·mL^−1^ for 4 h. Then, cultures were centrifuged and treated with a hypotonic solution (KCl 0.30 M) for 10 min, then cells were fixed with Carnoy’s solution (1 acetic acid:3 ethanol) and dropped onto slides. For the in situ hybridization, slides were firstly treated with pepsin/HCl (Sigma-Aldrich) (50 μg·mL^−1^) for 10 min at room temperature, washed twice with 1 × PBS for 5 min, fixed with formaldehyde 4%–PBS1×, and washed again with PBS (3 times, 5 min each). After that, slides were dehydrated in ethanol series (70%, 85%, and 100%, 2 min each) and air-dried. Then, hybridization with FITC-labeled pancentromeric and Cy3-labeled pantelomeric PNA probes (Applied Biosystems, Foster City, CA, USA) was performed. Shortly after, 10 μL of hybridizing solution with both PNA probes, 0.6 μM of the pancentromeric probe, and 0.4 μM of the pantelomeric probe were applied to the slides and covered with a coverslip. Cells and probes were denatured for 3 min at 80 °C and hybridized for 2 h at room temperature using a Vysis HYBrite Hybridization System (Abbott, Abbott Park, IL, USA). After hybridization, slides were washed twice, 15 min each, in 70% formamide–PBS1× and three times, 5 min each, in Tris-buffered saline solution, 1 × TBS/5% Tween20 (both from Sigma-Aldrich). Slides were then dehydrated with ethanol series and dried out, and 20 μL of the previously described DAPI solution were applied as counterstain and slides were observed on an epifluorescence microscope (AxioImager.Z2, Zeiss, Oberkochen, Germany).

### 2.4. Nanoparticles and Internalization

In the present study, three types of GNPs were used. Citric acid stabilized 50 nm GNPs (cat. BG-50, CD Bioparticles, Shirley, NY, USA), from now on, non-functionalized gold nanoparticles (NF-GNPs), commercially Tf functionalized gold nanoparticles (TfGNPs) (cat. GCT-50, CD Bioparticles) and anti-EpCam antibody functionalized gold particles (AntiEpCamGNPs) functionalized with a conjugation kit (cat. GCK-50, CD Bioparticles) according to the manufacturer’s protocol. To measure the ζ-potential, functionalized and non-functionalized GNPs were separately resuspended in water or in EMEM (anti-EpCamGNP) and McCoy’s 5A (TfGNP) culture mediums and sonicated for 5 min (Fisherbrand FB15047, Fisher Scientific, Germany) to achieve a monodispersed sample. The ζ-potential was then measured with a Zetasizer Nano ZS (Malvern Instruments, Malvern, UK).

Cell cultures of both cell lines were treated with 50 nm NF-GNPs. Moreover, Caco2 and SKBR3 cell cultures were treated with AntiEpCamGNPs and TfGNPs, respectively. In all cases, functionalized and non-functionalized GNPs, diluted in 1 × PBS, were added to the cell cultures at a final concentration of 7 × 10^5^ nanoparticles·mL^−1^. Nanoparticle internalization was evaluated 24 h after treatment of the cell cultures with a broadband confocal microscope (Leica TCS SP5, Leica Microsystems, Wetzlar, Germany) located at the Microscopy Service of the Universitat Autònoma de Barcelona. CellMask™ Deep Red (cat. C10046, Thermofisher, Waltham, MA, USA) was used to stain cell membranes and Hoechst 33342 (cat. H3570, Thermofisher) to stain cell nuclei. Staining was applied 5 min prior to observation. Due to their reflective nature, no additional staining was required to visualize GNPs.

### 2.5. Irradiation

Cell cultures were irradiated with gamma-rays 24 h after incubation in the absence or in the presence of nanoparticles (NF-GNPs for both cell lines and TfGNPs for SKBR3 cells and AntiEpCamGNPs for Caco2 cells). Irradiations were carried out at 3 Gy with a ^137^Cs irradiator (IBL437C, CIS Biointernational, GIF Yvette, France) located at the Technical Unit for Radiation Protection of the Universitat Autònoma de Barcelona. Dose rate was 5.02 Gy·min^−1^ and the energy peak was 662 keV. 

### 2.6. γ-H2AX Foci Detection

To evaluate kinetics of γ-H2AX foci, five different cultures for each cell line were used, one to determine the basal frequency and four at different postirradiation times: 30 min, 2, 4 and 24 h. At each time, three irradiations were carried out for each cell line, one without GNP treatment, another in the presence of NF-GNPs and another with functionalized GNPs (TfGNPs and AntiEpCamGNPs for SKBR3 and Caco2 cell lines, respectively). To detect γ-H2AX foci, an immunofluorescence staining was conducted following the same protocol described in the Section 2.2. In this case, using a specific primary mouse antibody for γ-H2AX (cat. ab26350, Abcam) and a goat anti-mouse Cy3 secondary antibody (cat. AP181C, Sigma-Aldrich). After this procedure, a couple of extra steps were taken to ensure proper microscopic analysis of radiation-induced foci. First, after the incubation with the secondary antibody, samples were again washed as described after the first incubation. Lastly, the slides were dehydrated with ethanol at 70%, 85% and 100% concentrations for 1 min each. Then, a DAPI counterstain was applied to the samples to allow single nucleus detection. 

γ-H2AX foci analyses were carried out using an automated scanning fluorescence microscope system (Metafer 4, Meta Systems, Altlussheim, Germany) and processed using the MetaCyte software, version 3.10.2, (Meta Systems, Medford, MA, USA) coupled to a motorized z-stage Zeiss Axio Imager.Z2 microscope (MetaSystems) The images were captured using a 63_PlanApo objective. The foci signals in the selected nuclei were captured using the SpOr filter (red channel). All the SpOr signals were acquired as a z-stack with a total of 10 focal planes and a z-step size of 0.35 μm between planes. A unique classifier was used to count foci in about 2000 nuclei for each experimental condition.

### 2.7. MTT Cell Viability Assay

Cells were seeded at 4 × 10^4^ cells·mL^−1^ concentration on a 96-well plate and cultured for 48 h before being treated with NF-GNPs, functionalized GNPs (TfGNPs and AntiEpCamGNPs for SKBR3 and Caco2 cell lines, respectively), or not treated at all in the case of our control cells. Twenty-four hours after GNP treatment, the culture medium was changed and then cells were irradiated at a 3 Gy dose. To determine the impact of treatments on cell viability, a 3-(4,5-dimethylthiazol-2-yl)-2,5-diphenyltetrazolium bromide tetrazolium reduction assay (MTT cell viability assay) was conducted following standard procedures 24 h after irradiation (cat. M2128, Sigma-Aldrich). Viability was calculated in relation to that observed in non-treated cells. Three replicates were carried out for each experimental condition.

### 2.8. Statistical Analysis

Statistical analysis was conducted with IBM SPSS Statistics 24. Normality was tested in all cases using a Kolmogorov–Smirnov test with the Lilliefors correction. Since all cases showed a non-normal distribution, the statistical test chosen to compare foci frequencies was the Mann–Whitney U test. Viability comparisons were carried out with ANOVA with Tukey’s multiple comparison test. *p* values lower than 0.05 were considered statistically significant.

## 3. Results

### 3.1. Receptor Expression

As expected, both cell lines expressed the cell membrane receptors (EpCam for Caco2 and TfR for SKBR3) described in the literature (Figure 1), confirming the validity of our choice for nanoparticle functionalization. 

### 3.2. Cytogenetic Characterization

The cytogenetic analysis by pancentromeric and pantelomeric fluorescence in situ hybridization (Figure 2) was carried out to know the modal karyotype of each cell line and if they show structural chromosome instability that could influence the further results with the foci analysis. The analyses were performed on 100 cells of each of the cell lines. Caco2 cells showed two modal karyotypes of 58 and 86 chromosomes whilst SKBR3 showed a modal karyotype of 76 chromosomes (Figure 3). Only one Caco2 cell showed a dicentric chromosome plus an acentric fragment, all the other chromosomes in all analyzed cells showed one centromere and two telomere signals. In the case of SKBR3, all the cells showed a stable dicentric chromosome, which is consistent with the data provided by the manufacturer. No cell showed another structural chromosome aberration detected by pancentromeric and pantelomeric labeling. 

These results indicate that instead of the variability in the number of chromosomes, the cell lines are stable regarding the formation of unstable structural chromosome aberrations.

### 3.3. Nanoparticle Internalization

In Figure 4 are shown the ζ-potentials of non-functionalized (NF-GNP) and functionalized GNPs (TfGNP and AntiEpCamGNP), measured in water and in culture media. As can be seen, in water NF-GNPs showed the most electronegative ζ-potential, a difference that disappeared in the culture medium where the ζ-potentials were less electronegative than in water. 

A qualitative assessment of GNP internalization through confocal microscopy imaging allowed us to detect certain differences between the cell lines (Figure 5 and Figure 6). As can be seen, for both cell lines the internalization of functionalized GNPs (TfGNP for SKBR3 and AntiEpCamGNP for Caco2 cells) is more visible than that for NF-GNPs. Moreover, it seems that SKBR3 cells showed more internalized functionalized GNPs than Caco2 cells (Figure 6). Finally, GNP aggregation was clearly visible for TfGNP in SKBR3 cells. Despite the different degrees of internalization between cell lines, both showed a common trait, that is, the inability of GNPs to enter the nucleus, thus remaining in the cytoplasm.

### 3.4. γ-H2AX Foci Induction and Kinetics

The cytogenetic characterization did not show structural chromosome aberrations, but due to the variability in the number of chromosomes, about 2000 cells were analyzed for each different postirradiation time and GNP treatment. Microscope images of γ-H2AX foci in Caco2 and SKBR3 cells are shown in Figure 7. In Caco2 cells (Figure 8), the higher frequencies of foci were observed 30 min after irradiation for all treatments: 64.3 ± 0.7 (mean ± SEM) without GNPs, 54.2 ± 0.8 with NF-GNPs, and 70.6 ± 0.7 with AntiEpCamGNPs. As postirradiation time went on, the frequencies of foci decreased to similar values to unirradiated cells. At any postirradiation time, the frequencies of foci in cells irradiated in the presence of AntiEpCamGNPs were significantly higher than for the other two treatments (*p* < 0.001).

For SKBR3 (Figure 9), the differences between treatments were even more noticeable. Cells irradiated in the absence of GNPs or in the presence of NF-GNPs showed a low foci frequency until two hours after irradiation (0.3 ± 1.0 and 1.3 ± 1.0 at 2 h, respectively). Then, an increase was observed at 4 h after irradiation (9.1 ± 0.5 and 17.4 ± 0.8, respectively), followed by a decrease to basal levels 24 h after irradiation. However, for cells irradiated in the presence of TfGNPs, the frequency of foci was maximum 30 min after irradiation, 36,1 ± 0.7, followed by a progressive decrease with postirradiation time but to values significantly higher than the basal ones at 24 h after irradiation, 8.9 ± 0.8. At all postirradiation times, the frequency of foci was significantly higher for cells irradiated in the presence of TfGNPs (*p* < 0.01). Table 1 shows the sensitization enhancement ratio (SER) for foci. As can be seen, cells irradiated in the presence of functionalized particles (either AntiEpCamGNPs or TfGNPs) showed ratios higher than one in all cases.

To know if the presence of GNPs could have any effect on the rate of foci disappearance with postirradiation time, foci decay with postirradiation time has been adjusted to a one-phase decay function, obtaining the constant of the decay rate and the foci half-life estimation for each experimental condition (Table 2). For Caco2 cells, the decay rate constant gradually decreases from cells irradiated without GNPs, to cells treated with non-functionalized GNPs and in cells treated with AntiEpCamGNPs. Moreover, the foci half-life was higher for cells irradiated in the presence of AntiEpCamGNPs. For the SKBR3 cells, because of the behavior in foci kinetics, this approach was only possible for cells irradiated in the presence of TfGNPs.

### 3.5. MTT Cell Viability Assay

Cell viability was measured by MTT analysis 24 h after irradiation (Figure 10). For both cell lines, without irradiation there were no significant differences in viability between cells grown without GNPs and cells grown with NF-GNPs or functionalized GNPs. However, for both cell lines the highest viability was observed for cells grown without GNPs. As expected, after irradiation the viability decreased significantly in all cases (*p* < 0.05). Moreover, for both cell lines the decrease in viability was more pronounced for cells irradiated in the presence of functionalized GNPs but was only significant for Caco2 cells (*p* < 0.05). After irradiation, SKBR3 cells always showed significantly lower viabilities than Caco2 cells (*p* < 0.05). The sensitization enhancement ratio for viability was also higher than one for functionalized particles (Table 1).

## 4. Discussion

The interest in improving radiotherapeutic treatments has led, in recent decades, to the development of new irradiation strategies focused on a more precise dose delivery to tumoral cells but also to radiosensitize tumoral tissues. One of these is the use of GNPs as a radiosensitizer agent, that was first evidenced experimentally in vivo by Hainfeld and collaborators [34]. In the present study, our aim was to evaluate the radiosensitizing effect of 50 nm GNPs specifically functionalized to be actively targeted to two tumoral cell lines. 

To measure the radiosensitizing effect of GNPs, different approaches such as viability and cytogenetic assays can be used. However, it should be noted that the sensitization enhancement ratio (SER) [16,35] can differ within the methodologies used, as had been previously described when the relative biological effectiveness of X-rays of different energies was checked [36,37,38,39,40,41,42,43,44,45,46,47]. 

The clonogenic assay has been used to measure the radioinduced cell death or loss of reproductive potential, though it has limitations such as the need for cell differentiation and the intrinsic variations in survivability between different cell types [48,49]. On the other hand, the cytogenetic approaches to the study of radiosensitivity are well known [50], with chromosomal damage and translocations being markers to test for radiation toxicity [51]. In the present study, we have used two methodologies to evaluate the radiosensitizing effect of GNPs: γ-H2AX foci detection as a genetic damage indicator and MTT as a cell viability assay. γ-H2AX is the phosphorylated variant of histone H2AX, forming discrete accumulations (foci) around the double strand breaks (DSBs) induced by the exposure to ionizing radiation [52,53,54]. γ-H2AX foci detection is a widely used technique for radiosensitivity assessment [55] because γ-H2AX foci can be detected almost immediately after the irradiation exposure, while other biomarkers such as chromosome aberrations require the cell to achieve the mitosis phase, thus yielding results only when the DNA repair mechanisms have already acted. Moreover, γ-H2AX foci can also be detected at different postirradiation time points to obtain a view on the kinetics of foci appearance and repair. Regarding the MTT cell viability assay, it allows a quick and easy sampling and quantification by spectrophotometry [56]. 

In our study, working with two different tumoral cell lines has helped us to achieve a better knowledge of the nanoparticles’ radiosensitizing effect in different cell models. First, to know if the cells used were genetically unstable, we evaluated the chromosome stability of both cell lines, and although the chromosome number was variable in both cell lines, no cells with chromosomes lacking telomere signals were observed, indicating no structural chromosome instability and hence no influence in the posterior foci counts. 

The radiosensitizing effect of GNPs is attributed, in part, to the increase in ROS production with irradiation, but in the absence of radiation, GNPs increase the intracellular ROS level [24,25,57,58]. It is known that ROS have a limited lifespan, so the radiosensitizing effect of GNPs should be more effective if nanoparticles are internalized by the cell. Our confocal microscope observations indicate that all GNPs are taken up by the cells, but the images obtained seem to indicate that functionalized GNPs (TfGNPs and AntiEpCamGNPs) were more effectively internalized by SKBR3 and Caco2 cells, respectively. It is well known that the surface charges of nanoparticles influence their internalization by the cells, with highly charged nanoparticles (positively or negatively) being taken up more by cells than neutral ones. It has also been described that functionalization of polystyrene macroparticles to make them positively charged significantly increases their internalization by HeLa cells [59]. We observed that the ζ-potential of all GNPs (functionalized and non-functionalized) becomes less electronegative when the ζ-potential was measured in culture medium, unless in water. Moreover, the differences observed in water disappeared in cell culture medium. This could be due to a protein corona formation, with serum proteins, when particles are in culture medium. This modification has been suggested to be responsible for the decrease in particle uptake by the cells [60,61], but it has also been described that the impact can vary between cell types [62]. In any case, for an accurate quantitative assessment of GNP uptake by the cells, alternative methods such as those based on inductively coupled plasma mass spectrometry (ICP-MS) should be applied.

In our study, we chose nanoparticles of 50 nm in diameter because previous studies indicated an increased cell uptake for GNPs of 50 nm in diameter in relation to 14 nm and 74 nm [63] and it has been correlated with a higher radiosensitization effect [22]. On the other hand, it seems that smaller nanoparticles allow the deposition of high doses in their vicinity than bigger nanoparticles because of the increase in the surface to volume ratio [23]. However, because different studies have been carried out with different nanoparticle sizes, it is difficult to establish a unique optimum size of nanoparticles to increase cell uptake. 

Another factor, different to nanoparticles properties, to consider in studies of GNP radiosensitizing effect, is the intrinsic uptake capacities and intrinsic radiosensitivity of the cell lines. Not all cell types show a similar nano- or microparticle uptake capacity. SKBR3 cells can phagocytose 3 µm polysilicon–chromium–gold microparticles [64], 3 µm polystyrene microparticles functionalized with polyethyleneimine [61] and 1 µm and 3 µm polystyrene Tf-functionalized microparticles [65] without affecting viability. Caco2 cells can uptake functionalized polystyrene nanoparticles of 50 nm in diameter, but not of 100 nm or more [66], and were also unable to uptake TiO2 nanoparticles [67]. Actively targeting using anti-EpCam antibodies is an effective method to target Caco2 cells with magnetic and gold nanoparticles functionalized with these antibodies [68,69]. Other studies indicate that the capability of Caco2 cells to uptake nanoparticles depends on their differentiated status, with only the undifferentiated cells being able to phagocytose nanoparticles [70,71]. Considering these results, one objective of our work was to determine if actively targeted GNPs could increase the radiosensitizing effect. 

Regarding the effect of GNPs in SKBR3 and Caco2 cell cultures without irradiation, we observed that the presence of NF-GNPs or functionalized GNPs did not significantly change the basal frequency of γ-H2AX foci or the cell viability when compared to cells grown without GNPs. This does not mean that the presence of GNPs does not have any effect on cells, because a slight decrease in cell viability, though not statistically significant, was observed when GNPs (either non-functionalized or functionalized) were present in the cell culture. This agrees with studies describing an increase in ROS by the presence of GNPs without irradiation [24,25,57,58] that could have a slight effect on cell viability.

After irradiation, the γ-H2AX foci frequency was evaluated at four different postirradiation times (30 min, 2, 4 and 24 h). For Caco2 cells, significant increases in foci frequencies were observed for cells irradiated in the presence of AntiEpCamGNPs when compared with cells irradiated in the presence of NF-GNPs or in the absence of GNPs. For these cells, the sensitization enhancement ratios (SERs) for foci obtained for AntiEpCamGNPs when compared with NF-GNPs or with irradiation without GNPs were, in general, close to the values described in the literature (reviewed in [15,16]). Moreover, the residual DNA damage was higher when irradiation was carried out in the presence of GNPs, either functionalized or non-functionalized. For SKBR3, the γ-H2AX foci kinetics showed a particular behavior when compared to Caco2 cells. The maximum foci frequency in those cells treated without GNPs or with NF-GNPs was delayed to 4 h postirradiation, while for cells treated with TfGNPs the maximum frequency of foci was observed 30 min after irradiation. It is difficult to explain the delayed appearance of foci in cells irradiated without GNPs or in the presence of NF-GNPs. A delayed appearance of γ-H2AX foci in different cell lines has been described [72], usually being related to an ATM deficiency, though the exact mechanism remains unclear. Despite this particular behavior, for all postirradiation times the frequency of foci in cells irradiated in the presence of TfGNPs was significantly higher than those for cells irradiated with NF-GNPs or in the absence of GNPs. For SKBR3 cells, TfGNPs always showed an elevated radiosensitizing effect when compared with NF-GNPs or no GNPs, but in this case the SER values were too high because of the strange behavior of foci frequencies for irradiations with NF-GNPs or without GNPs. In addition, it has been described that GNPs can interfere with the DNA repair machinery [24]. In the present study, and with Caco2 cells, the half-life of foci was higher in those cells treated with GNPs, being the highest in those treated with functionalized GNPs, indicating that GNPs would have this dual effect on cells irradiated in their presence, an increase in the DNA damage production but also a reduction in the DNA repair efficiency. Anyway, the overall results indicate that actively targeting GNPs has a radiosensitizing effect in both cell lines. This result is also supported by cell viability analyses. The mean viability of Caco2 cells irradiated in the presence of AntiEpCamGNPs (79.9% of viable cells) was lower than for cells irradiated in the presence of NF-GNPs (85.3%) or in the absence of GNPs (88. 7%). For SKBR3 cells, the mean viabilities were 59% for TfGNPs, 60% for NF-GNPs and 66.1% for cells irradiated in the absence of GNPs. These results are similar to those reported with gamma-rays [21,22]. 

Finally, when foci frequencies of both cell lines are compared, in all experimental conditions lower frequencies were observed in SKBR3 cells. This could be related to an elevated resistance to oxidative stress, as described for cells from breast cancer patients [73]. However, instead of the low levels of foci for SKBR3 cells when compared with Caco2 cells, their viability is clearly lower, indicating that Caco2 is a very radioresistant cell line, probably because of its lower expression of 53BP1.

Although high Z metallic nanoparticles are currently the most used for radiosensitizing, other non-metallic nanoparticles are showing promising results. In a study by Yamaguchi et al. [74], silicon oxide nanoparticles coated with polyamidoamine have yielded a significant radiosensitizing effect on SKBR3 cells, due to their overexpression of HER2. Moreover, nanoparticle functionalization with therapeutic molecules, specifically targeted to a cell type, can be a powerful tool to fight against cancer. Using GNPs, a 43-fold decrease in viability was observed when SKBR3 cells were irradiated in the presence of trastuzumab-functionalized GNPs when compared with the same cells irradiated in the presence of non-functionalized GNPs, indicating the importance of active targeting [75].

Despite the radiosensitizing effect of gold nanoparticles previously described [76], the results of the present study led us to question the universality of their application. Working with only two cell lines has already shown how much the radiosensitizing effect of GNPs can vary due to the cell lines’ intrinsic radiosensitivity but also by the capability to take up GNPs. Besides the differences between cells, the choice of targeting molecules and nanoparticle composition, size and shape are also key factors to consider in further studies.

## 5. Conclusions

Even with the differences in the results of the present study between the two cell lines, gold nanoparticles remain an interesting radiosensitizing agent and a good candidate for future in vivo application in the treatment of certain types of tumors. Active targeting of GNPs has demonstrated its effectiveness to increase the genetic damage and to decrease the cell viability. This GNP radiosensitizer effect should be explored in further studies to find more effective radiation therapies against different tumor types.

## Figures and Tables

**Figure 1 biology-11-01193-f001:**
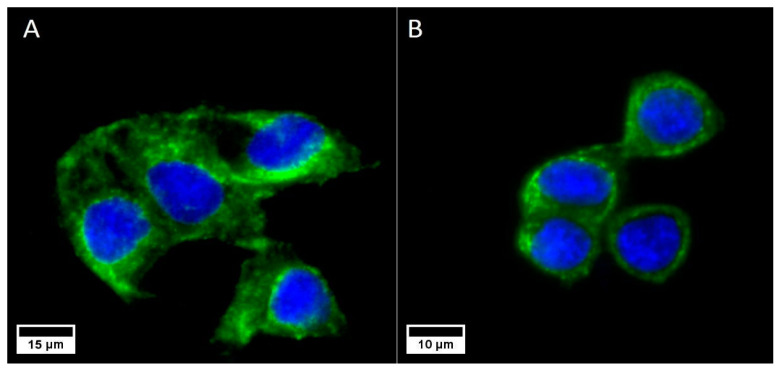
Caco2 cells expressing the EpCam (**A**) and SKBR3 cells expressing the TfR (**B**) receptors. In both cases, the nuclei are marked in blue (DAPI) and the cell membrane receptors are marked in green (Alexa Fluor^®^ 488, Abcam).

**Figure 2 biology-11-01193-f002:**
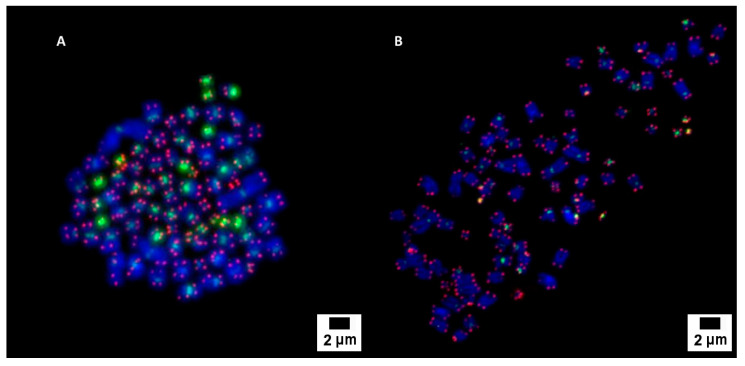
Pancentromeric–pantelomeric fluorescence in situ hybridization-stained metaphases of Caco2 (**A**) and SKBR3 (**B**) cells. Green signals correspond to centromeres, labeled with FITC; and red signals to telomeres, labeled with Cy3.

**Figure 3 biology-11-01193-f003:**
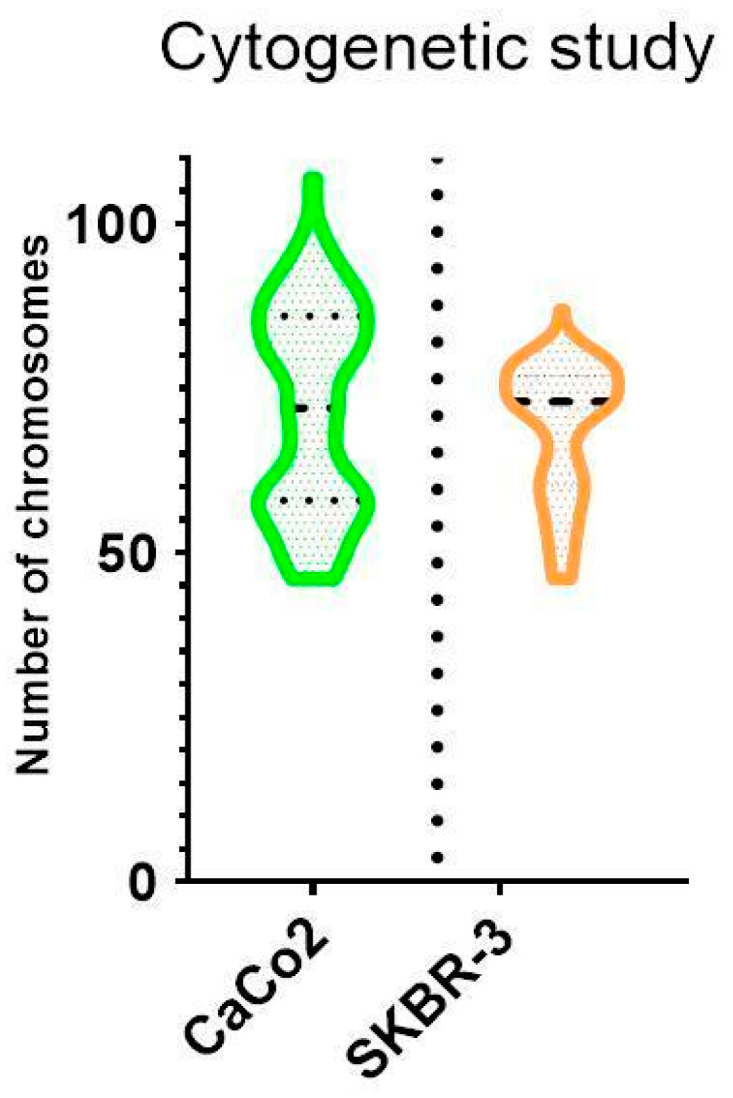
Violin plot showing chromosome distribution of both cell lines. The mean number of chromosomes for Caco2 was 72, although two modal karyotypes of 58 and 86 chromosomes were observed. A unique modal karyotype of 76 chromosomes for SKBR3 was observed.

**Figure 4 biology-11-01193-f004:**
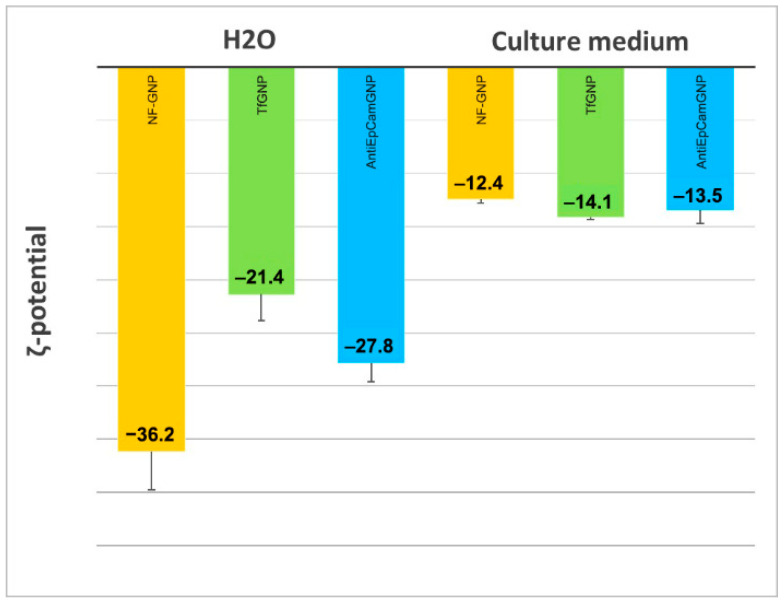
ζ-potential of non-functionalized (NF-GNP) and functionalized GNPs (TfGNP and AntiEpCamGNP), measured in water and in EMEM (AntiEpCamGNP) and McCoy’s 5A (TfGNP) culture media.

**Figure 5 biology-11-01193-f005:**
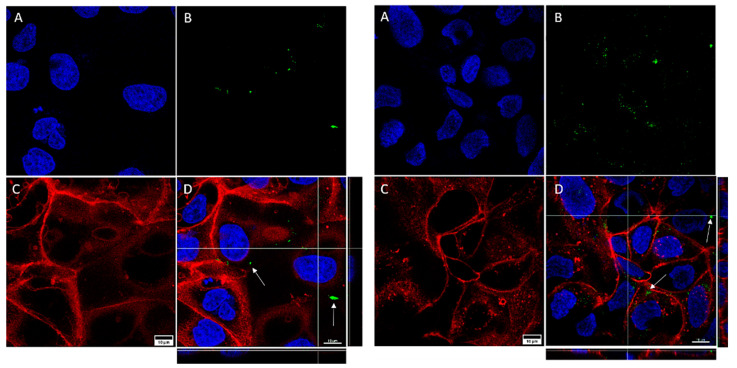
Confocal microscopy captures of Caco2 cells. The images show cells treated with NF-GNPs (left) and cells treated with anti-EpCam-functionalized GNPs (right). Nuclei (**A**) are marked blue (Hoechst 33342), plasma membranes (**C**) are marked red (CellMask™ Deep Red). Nanoparticles (**B**) are visualized as green dots due to their reflection being artificially assigned a green color by the microscope’s software. Merged images (**D**), and their orthogonal projections of the z-stack reconstructions (right and bottom), where GNPs (white arrows) can be seen.

**Figure 6 biology-11-01193-f006:**
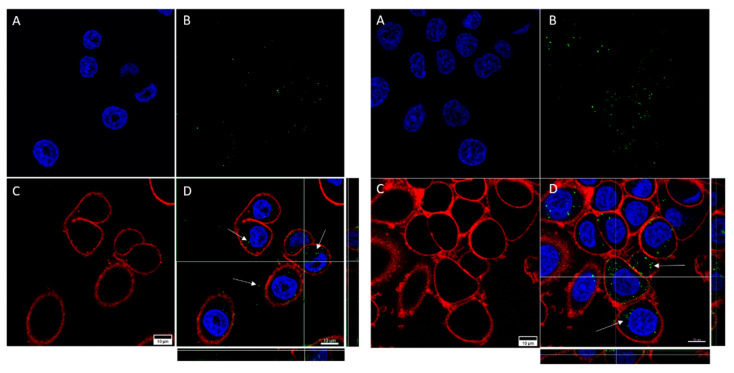
Confocal microscopy captures of SKBR3 cells. The images show cells treated with NF-GNPs (left) and cells treated with Tf-functionalized GNPs (right). Nuclei (**A**) are marked blue (Hoechst 33342), cytoplasmatic membranes (**C**) are marked red (CellMask™ Deep Red). Nanoparticles (**B**) are visualized as green dots due to their reflection being artificially assigned a green color by the microscope’s software. Merged images (**D**), and their orthogonal projections of the z-stack reconstructions (right and bottom), where GNPs (white arrows) can be seen.

**Figure 7 biology-11-01193-f007:**
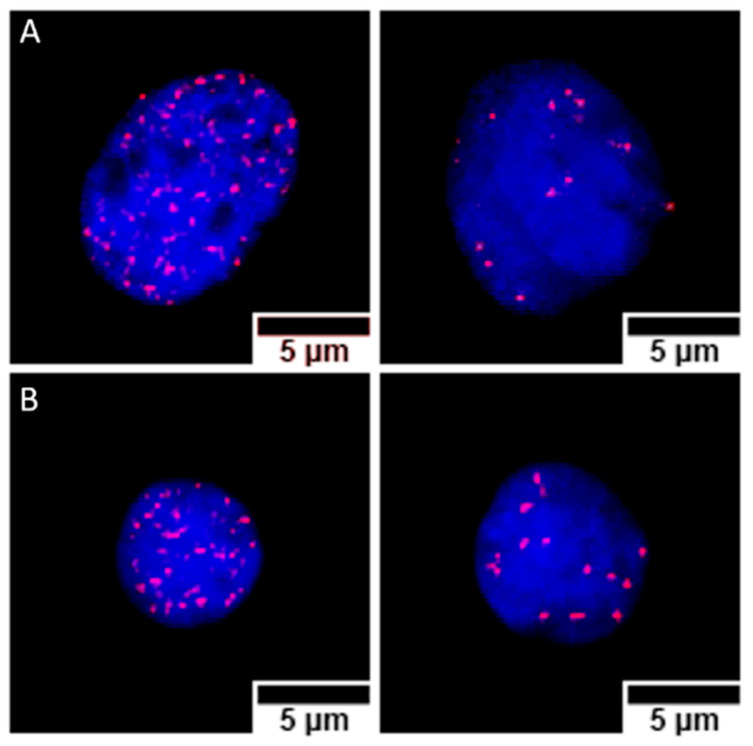
Nuclei from Caco2 (**A**) and SKBR3 (**B**) cells, stained with DAPI (blue), containing γ-H2AX foci immunostained with cyanine 3 (Cy3, red). For both cell lines, nuclei with high (left) and low (right) foci counts are shown. The images correspond to samples prepared 30 min (high foci count) and 4 h (low foci count) after irradiation, treated with anti-EpCam-functionalized GNPs (Caco2) or transferrin-functionalized GNPs (SKBR3).

**Figure 8 biology-11-01193-f008:**
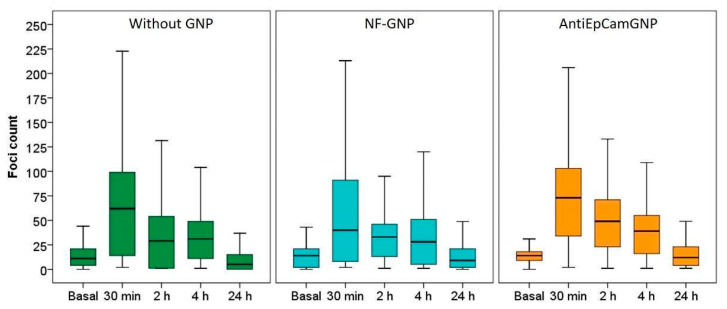
Box plots of the number of foci per cell in Caco2 cells before (basal) and at different times (30 min, 2, 4 and 24 h) after irradiation, in the absence of GNPs (Without GNP) or in the presence of non-functionalized GNPs (NF-GNP) or anti-EpCam-functionalized GNPs (AntiEpCamGNP). Horizontal lines inside boxes indicate the median.

**Figure 9 biology-11-01193-f009:**
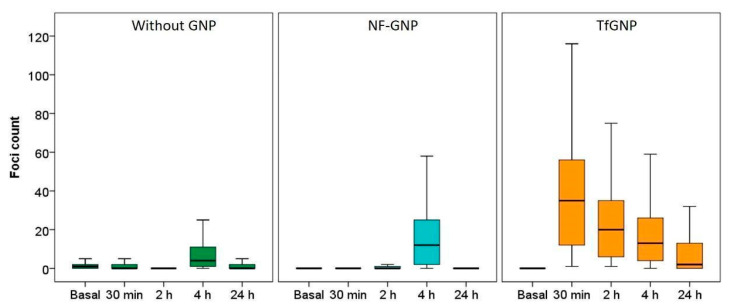
Box plots of the number of foci per cell in SKBR3 cells before (basal) and at different times (30 min, 2, 4 and 24 h) after irradiation, in the absence of GNPs (Without GNP) or in the presence of non-functionalized GNPs (NF-GNP) or transferrin-functionalized GNPs (TfGNP). Horizontal lines inside boxes indicate the median.

**Figure 10 biology-11-01193-f010:**
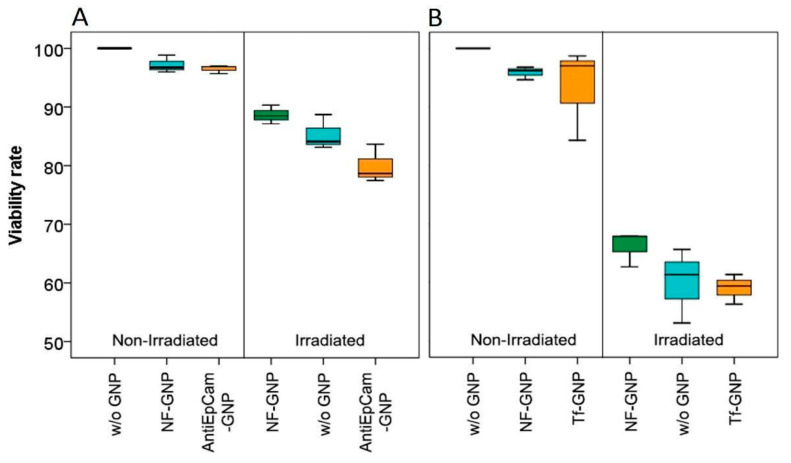
Viability rates of Caco2 (**A**) and SKBR3 (**B**) cells, left, non-irradiated and right, 24 h after irradiation. From left to right, the boxes mark the viability of cells in the absence of GNPs (w/o GNP), in the presence of non-functionalized GNPs (NF-GNP) and in the presence of functionalized GNPs (AntiEpCam-GNP and Tf-GNP, respectively).

**Table 1 biology-11-01193-t001:** Sensitization enhancement ratio (SER) for foci frequencies and viability. Caco2 and SKBR3 cells were irradiated at 3Gy in the presence of functionalized GNPs (AntiEpCamGNP and TfGNP, respectively) and results were compared with those after irradiation in the absence of GNPs (w/o GNP) or in the presence of non-functionalized GNPs (NF-GNP).

	AntiEpCamGNP vs.		TfGNP vs.	
Postirradiation Time	w/o GNP	NF-GNP	w/o GNP	NF-GNP
	Foci SER		Foci SER	
30 min	1.10	1.30	-	-
2 h	1.46	1.55	-	-
4 h	1.20	1.18	2.05	1.07
24 h	1.87	1.18	4.98	22.57
	Viability SER		Viability SER	
	1.11	1.07	1.12	1.02

**Table 2 biology-11-01193-t002:** Constant of the decay rate (K) and foci half-life for cells irradiated at 3 Gy in the three experimental conditions, in the absence of GNPs (w/o GNPs), in the presence of non-functionalized GNPs (NF-GNP) and in the presence of specifically targeted GNPs (AntiEpCamGNP and TfGNP for Caco2 and SKBR3 cells, respectively).

	Irradiation Conditions	K (h^−1^)	Foci Half-Life (h)
CaCo2	w/o GNPs	0.35	2.00
	NFGNP	0.32	2.17
	AntiEpCamGNP	0.28	2.44
SKBR3	w/o GNPs	-	-
	NFGNP	-	-
	TfGNP	0.30	2.31

## Data Availability

The data presented in this study are available on request from the corresponding author.

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
