# Peer review of "Differential Radiosensitizing Effect of 50 nm Gold Nanoparticles in Two Cancer Cell Lines"

_biology, 2022, doi:10.3390/biology11081193_

Round 1
Reviewer 1 Report
The authors have extensively reviewed and reformulated the article in different sections. I do value their effort and I consider this version much improved.
However, the comment about icp-ms is not fully answered. This reviewer suggests that alternative methods to assess NPs internalization in a quantitative way should be mentioned in the discussion, for the sake of completeness.
"This reviewer finds that alternative methods such as ICP-MS should be performed to quantitatively evaluate the internalization of both types of aunps." Unfortunately, this is the only point addressed by the reviewer that we cannot respond to.
Author Response
Response
In agreement to the reviewer’s comment, we have added the last sentence of this paragraph in the discussion section, page 12 line 417.
The radiosensitizing effect of GNPs is attributed, in part, to the increase of ROS production when irradiation is done in their presence, but also in absence of radiation, GNPs increase the intracellular ROS level [24; 25; 57; 58]. It is known that ROS have a limited lifespan, so the radiosensitizing effect of GNPs should be more effective if nanoparticles are internalized by the cell. Our confocal microscope observations indicate that all GNPs are uptaked by the cells, but the images obtained seems to indicate that functionalized GNPs (TfGNP and AntiEpCamGNP) were more effectively internalized by SKBR3 and Caco2 cells respectively. It is well known that the surface charges of nanoparticles influence their internalization by the cells, being highly charged nanoparticles (positively or negatively) more uptaked by cells than neutral ones. It has also been described that functionalization of polystyrene macroparticles to make them positively charged increases significantly their internalization by HeLa cells [59]. We observed that the ζ-potential of all GNPs (functionalized and non-functionalized) become less electronegative when the ζ-potential was measured in culture medium unless in water. Moreover, the differences observed in water disappeared in cell culture medium. This could be due to a protein corona formation, with serum proteins, when particles are in culture medium. This modification has been suggested to be the responsible of the decrease in particles uptake by the cells [60, 61], but it has also been described that the impact can vary between cell types [62]. In any case, for an accurate quantitative assessment of GNPs uptake by the cells, alternative methods such as those based on inductively coupled plasma mass spectrometry (ICP-MS) should be applied.

Reviewer 2 Report
Manuscript ID : Biology-1814266
Article: "Differential radiosensitizing effect of 50 nm gold nanoparticles in two cancer cell lines"
General comment : Manuscript is well written, presents interesting results.
Minor comments :
- Figure 2 : Could you add a scale bar on these images ?
- Figure 8 : The foci count you reported is a number of foci/cell or a total number of foci on the image you analyzed ? I'm a little confused.
Major comment :
- Result (section 3.4.) : you claim "At any postirradiation time, the frequencies of foci in cells irradiated in the presence of AntiEpCamGNP were significantly higher than for the other two treatments (p<0.001)."
DNA repair is a dynamic process were 2 processes is simultaneously happening : DNA damage recognition by DDR sensor protein and the repair of DNA damage. In your experimental design, you investigated this by analyzing the number of DNA damage at specific timepoints post-irradiation. You showed a higher foci frequency for all timepoints with your GNP, so I'm agree with you that it seems to mean a higher DNA damage formation. However, what about the impact of GNP on the DNA repair ?
Some studies evidenced the ability of GNP to inhibit/slow down the DNA repair. This will lead to a higher amount of DNA damage at specific time-point compared to conditions without GNP. I think you have to analyze your data using mathematical formalism to distinguish the DNA damage production and the kinetic of DNA repair in order to draw stronger conclusions on this. In their recent guidelines, Costes' Lab recommended a mathematical formalism to reach this goal (equation 3 ; https://doi.org/10.1093/narcan/zcab046).). Apply this to your experimental data and put the different fitting parameters in a table to discuss it.
Author Response
General comment : Manuscript is well written, presents interesting results
Minor comments :
Figure 2 : Could you add a scale bar on these images ?
Scale bars have been added to figure 2
Figure 8 : The foci count you reported is a number of foci/cell or a total number of foci on the image you analyzed ? I'm a little confused.
The reviewer is right, in figures 8 and 9 are represented the number of foci per cell in each of the 2000 cells analyzed for each time and experimental condition. This data has been converted to a box plot. We have modified the figure caption as follows:
Figure 8. Foci frequencies Box plots of the number of foci per cell in Caco2 cells before (basal) and at different times (30 min, 2, 4 and 24 h) after irradiation, in the absence of GNPs (Without GNP) or in the presence of non-functionalized GNPs (NF-GNP) or AntiEpCam-functionalized GNPs (AntiEpCamGNP). Horizontal lines inside boxes indicate the median.
Figure 9. Foci frequenciesBox plots of the number of foci per cell in SKBR3 cells before (basal) and at different times (30 min, 2, 4 and 24 h) after irradiation, in the absence of GNPs (Without GNP) or in the presence of non-functionalized GNPs (NF-GNP) or Transferrin-functionalized GNPs (TfGNP). Horizontal lines inside boxes indicate the median.
Major comment :
Result (section 3.4.): you claim "At any postirradiation time, the frequencies of foci in cells irradiated in the presence of AntiEpCamGNP were significantly higher than for the other two treatments (p<0.001)."
DNA repair is a dynamic process were 2 processes is simultaneously happening : DNA damage recognition by DDR sensor protein and the repair of DNA damage. In your experimental design, you investigated this by analyzing the number of DNA damage at specific timepoints post-irradiation. You showed a higher foci frequency for all timepoints with your GNP, so I’m agree with you that it seems to mean a higher DNA damage formation. However, what about the impact of GNP on the DNA repair ?
Some studies evidenced the ability of GNP to inhibit/slow down the DNA repair. This will lead to a higher amount of DNA damage at specific time-point compared to conditions without GNP. I think you have to analyze your data using mathematical formalism to distinguish the DNA damage production and the kinetic of DNA repair in order to draw stronger conclusions on this. In their recent guidelines, Costes' Lab recommended a mathematical formalism to reach this goal (equation 3 ; https://doi.org/10.1093/narcan/zcab046).). Apply this to your experimental data and put the different fitting parameters in a table to discuss it.
We agree with the reviewer’s comment indicating that the presence of GNPs may result not only in an increase of the radiation-induced DSBs, but also in a reduction of DSBs repair efficiency. However, because we have not analyzed any point before 30 minutes, we believe that the k1 values will not be representative of the RIF formation from DSB. In any case, according to the reviewer comment we have applied the exponential one phase decay model to know the kinetics of RIF disappearance with post-irradiation time as well as the RIF half-life. We used this method previously to compare these parameters between two cell lines with different radiosensitivity (Borras et al, Scientific Reports | 6:27043 | DOI: 10.1038/srep27043)
To include this effect in the manuscript we have added a paragraph and a table in the results section (page 10 line 330).
“To know if the presence of GNPs could have any effect on the rate of foci disappearance with post-irradiation time, foci decay with postirradiation time has been adjusted to a one-phase decay function, obtaining the constant of the decay rate and the foci half-life estimation for each experimental condition (table 2). For Caco2 cells, the decay rate constant gradually decreases from cells irradiated without GNPs, to cells treated with non-functionalized GNPs and in cells treated with AntiEpCamGNP. Moreover, the foci half-life was higher for cells irradiated in the presence of AntiEpCamGNP. For the SKBR3 cells, because of the behavior in foci kinetics, this approach was only possible for cells irradiated in the presence of TfGNP.”
Table 2. Constant of the decay rate (K) and foci half-life for cells irradiated at 3 Gy in the three experimental conditions, in absence of GNPs (w/o GNPs) in the presence of non-functionalized GNPs (NF-GNP) and in the presence of specifically targeted GNPs (AntiEpCamGNP and TfGNP for Caco2 and SKBR3 cells respectively).
|
Irradiation conditions |
K (h-1) |
Foci half-life (h) |
CaCo2 |
w/o GNPs |
0,35 |
2,00 |
NFGNP |
0,32 |
2,17 |
|
AntiEpCamGNP |
0,28 |
2,44 |
|
SKBR3 |
w/o GNPs |
- |
- |
NFGNP |
- |
- |
|
TfGNP |
0,30 |
2,31 |
We have also included the following paragraph in the discussion section, page 13 line 471.
“In addition, it has been described that GNPs can interfere with the DNA repair machinery (26). In the present study, and with Caco2 cells, the half-life of foci was higher in those cells treated with GNPs, being the highest in those treated with functionalized GNPs. Indicating that GNPs would have this dual effect on cells irradiated in their presence, an increase in the DNA damage production but also a reduction of the DNA repair efficiency.

Reviewer 3 Report
I carefully read the the response to my comments and the new submitted manuscript.
I´m still not convinced that the study significantly improved. Here some of my concerns:
1) The aim of this work is to compare the radiosensitizing effect of non-functionalized and functionalized AuNPs. A sole qualitative measurement of the AuNPs by the different cell lines is not enough. Additionally the difference showed in the fluorescence microscope images are not convincing.
2) The difference in AuNP uptake between non-funtionalized and functionalized nanoparticles is minimal. I don´t understand why the authors needed to target specific membrane receptors. Here it would be best to show if there is a difference between the cancer and the according non-tumorigenic cells, but this study lacks this information and is therefore incomplete.
3) The author show the zeta potential measurements of the AuNPs. The decrease of the zeta potential is explained by building an protein corona. There is no measurement of the hydrodynamic diameter and the potential formation of agglomerates, which influences the uptake of the AuNPs.
4) There is no proof in the manuscript that the functionalization of the AuNP was successful
I could not recommened this manuscript for publication.
Author Response
Comments and Suggestions for Authors
I carefully read the response to my comments and the new submitted manuscript.
I´m still not convinced that the study significantly improved. Here some of my concerns:
1) The aim of this work is to compare the radiosensitizing effect of non-functionalized and functionalized AuNPs. A sole qualitative measurement of the AuNPs by the different cell lines is not enough. Additionally the difference showed in the fluorescence microscope images are not convincing.
We agree with the reviewer that we have not done quantitative analysis of AuNPs uptake. However, our data indicate that the presence of AuNPs during irradiation increases the frequency of foci, and decreases the cell viability when compared with cells irradiated in the absence of AuNPs. Moreover, these effects were more pronounced when AuNPs were specifically targeted. This is not a direct evidence but an indirect evidence of an increase in the radiosensitizing effect when AuNPs are specifically targeted.
2) The difference in AuNP uptake between non-funtionalized and functionalized nanoparticles is minimal. I don´t understand why the authors needed to target specific membrane receptors. Here it would be best to show if there is a difference between the cancer and the according non-tumorigenic cells, but this study lacks this information and is therefore incomplete.
As the reviewer indicates in point 1, we have not quantified AuNPs uptake and we cannot ensure if uptake differences are more or less important. However, because the results obtained showed a slight, but significant, radiosensitizing effect of specifically targeted AuNPs, it pointed out to indirect evidence of a slightly higher uptake of functionalized AuNPs. For this reason, we only indicate that we found “certain differences” in a qualitative analysis.
In relation to the need of specific targeting, our purpose is to evaluate the impact of irradiation when it is done in the presence of specifically targeted gold nanoparticles. For this reason we also performed the same analyses on cells irradiated in the absence of AuNPs and in the presence of non-functionalized AuNPs. This is necessary to detect if a radiosensitizing effect exists or not when AuNPs are specifically targeted to membrane receptors. Because both AuNPs (functionalized and non-functionalized) were observed inside cells, and the impact of irradiation was higher when AuNPs were specifically targeted there is a qualitative indirect evidence of a slight uptake of functionalized AuNPs.
3) The authors show the zeta potential measurements of the AuNPs. The decrease of the zeta potential is explained by building a protein corona. There is no measurement of the hydrodynamic diameter and the potential formation of agglomerates, which influences the uptake of the AuNPs.
We agree with the reviewer that changes in the hydrodynamic diameter can, undoubtedly, influence the uptake of AuNPs. However, in other studies of our group, with microparticles (references 59 and 61), we did not measure the changes in the hydrodynamic diameter. In the present study the formation of a protein corona probably affects to non-functionalized and functionalized AuNPs but we cannot know the extent of this surface modification because unfortunately we are not able to carry out measurements of hydrodynamic diameter.
4) There is no proof in the manuscript that the functionalization of the AuNP was successful
As indicated in response to points 1 and 2, we believe that there also exist indirect evidence of a successful functionalization. If not, how can we explain the observed radiosensitization effect? We agree that the magnitude of this effect is not as high as we expected, but this does not mean that didn’t exist.
I could not recommend this manuscript for publication.
In any case, we thank the reviewer for their comments because they will be very helpful for us in the future.

Reviewer 4 Report
The authors have considered all the points commented by this referee. The manuscript has improved considerably and deserves to be published.
Author Response
The are no responses.
Round 2
Reviewer 2 Report
Thanks to the authors for this nice revised version
This manuscript is a resubmission of an earlier submission. The following is a list of the peer review reports and author responses from that submission.
Round 1
Reviewer 1 Report
This reviewer suggests an exhaustive revision of the style of the manuscript. This reviewer considers that some points, particularly in terms of description of the results, also need to be addressed. Some examples are enumerated below.
Overall, improvement of the manuscript and some essential additional studies need to be performed, before it is suitable for publication.
In accordance with the suggestions of the reviewers and to address all comments made, the manuscript has been modified in its entirety. In this letter, the authors indicate the modifications done in the manuscript to respond the reviewer's comments and suggestions
General Comments
The authors compare 2 different cancer cell lines with different radiosensitivity. This is a highly interesting approach and could contribute with relevant information to the field. However the study is performed with 2 different types of functionalized gold nanoparticles (Aunps). This brings an additional factor into the analysis, which is the degree of interaction and internalization with the targeted receptors (and also the degree of expression of the 2 receptors in the respective cell lines). Further understanding of the behaviour of the aunps and their internalization is needed.
We agree with the reviewer’s comment that understanding how functionalized AuNps (GNPs) are internalized by different cell lines depends on the intrinsic characteristics of the cells, as well as characteristics of the functionalized particles themselves. In the discussion section we have discussed different factors that influence GNPs internalization. We believe that these paragraphs give a better understanding of this issue.
Discussion section, pages 13-14.
“Another factor, different to nanoparticles properties, to consider in studies of GNPs radiosensitizing effect, is the intrinsic uptake capacities and intrinsic radiosensitivity of the cell lines. Not all cell types show a similar nano or microparticles uptake capacity. SKBR3 cells can phagocyte 3μm Polysilicon-chromium-gold microparticles [64], 3μm Polystyrene microparticles functionalized with Polyethyleneimine [61] and 1μm and 3μm Polystyrene Tf functionalized microparticles [65] without affectation on viability. Caco2 cells can uptake functionalized polystyrene nanoparticles of 50 nm in diameter, but not of 100 nm or more [66], and were also unable to uptake TiO2 nanoparticles [67]. Actively targeting using AntiEpCam antibodies is an effective method to target Caco2 cells with magnetic and gold nanoparticles functionalized with these antibodies [68, 69]. Other studies indicate that the capability of Caco2 cells to uptake nanoparticles depends on their differentiated status, being only the undifferentiated cells able to phagocyte nanoparticles [70, 71]. Considering these results one objective of our work was to determine if actively targeted GNPs could increase the radiosensitizing effect.”
There seems to be some confusion between survival (line 24) and viability (line 84), mentioned then all throughout the MS. The MTT assay is not a survival detection method, as authors clarify in the discussion (line 310 and following).
This was a mistake, as the reviewer indicates, MTT is an assay for testing cell viability. This has been clarified in the entire manuscript, in the materials methods 2.7 section, in the results 3.5 section, and in the discussion.
Some examples of points to specifically address/ correct
Methods section
Entries should be organized uniformly- some are methods, other are the objective of the method (p.ex Receptor expression)
The entries in the methods section have been modified as follows:
Cell lines; Receptor expression; Cytogenetic characterization; Nanoparticles and internalization; Irradiation; ᵞ-H2AX foci detection; MTT cell viability assay and statistical analysis.
Entries (2.4 and 2.5) regarding the evaluation of AuNPs need clarification:
- 24h after treatment for microscopy and irradiation – is it 24h of exposure to the nps?
Confocal analysis was done after 24 hours incubation with different GNPs. This is now indicated in the second paragraph of 2.4 section:
“Cell cultures of both cell lines were treated with 50 nm NF-GNP. Moreover, CaCo2 and SKBR3 cell cultures were treated with AntiEpCamGNP and TfGNP respectively. In all cases, functionalized and non-functionalized GNPs, diluted in 1X PBS, were added to the cell cultures at a final concentration of 7x105 nanoparticles·mL-1. Nanoparticle internalization was evaluated 24 h after treatment of the cell cultures with a broadband confocal microscope (Leica TCS SP5, Leica Microsystems, Wetzlar, Germany) located at the Microscopy Service of the Universitat Autònoma de Barcelona. CellMask™ Deep Red (cat. C10046, Thermofisher, Waltham, Massachusetts, USA) was used to stain cell membranes and Hoechst 33342 (cat. H3570, Thermofisher) to stain cell nuclei. Staining was applied 5 minutes prior to observation. Due to their reflective nature, no additional staining was required to visualize GNPs.”
Irradiation was also done after 24h GNPs treatment, this is now indicated in the first sentence of the 2.5 section:
“Cell cultures were irradiated with gamma-rays 24 h after incubation in the absence or in the presence of nanoparticles (NF-GNP for both cell lines and TfGNP for SKBR3 cells and anti-EpCamGNP for Caco2 cells). Irradiations were done at 3 Gy with a 137Cs irradiator (IBL437C, CIS Biointernational, GIF Yvette, France) located at the Technical Unit for Radiation Protection of the Universitat Autònoma de Barcelona. Dose rate was 5.02 Gy·min−1 and the energy peak was 662 keV.”
Mtt – not clear the methodological approach – AunPS incubation ofr 24h + irradiation + 24h later plating + xh? when was the MTT assay performed?
The MTT cell viability assay is now better explained in section 2.7 section:
“Cells were seeded at a 4x104 cells·mL-1 concentration on a 96-well plate and cultured for 48 h before being treated with NF-GNP, functionalized GNPs (TfGNP and anti-EpCamGNP for SKBR3 and Caco2 cell lines respectively), or not treated at all in the case of our control cells. 24 hours after GNPs treatment, culture medium was changed and then cells were irradiated at a 3 Gy dose. To determine the impact of treatments on cell viability, a 3-(4,5-dimethylthiazol-2-yl)-2,5-diphenyltetrazolium bromide tetrazolium reduction assay (MTT cell viability assay) was conducted following standard procedures 24 h after irradiation (cat. M2128, Sigma-Aldrich). Viability was calculated in relation to that observed in non-treated cells. Three replicates were done for each experimental condition.”
2.4
Is its mentioned:
Gold nanoparticles of 50 nm diameter (CD Bioparticles, Shirley, NY, USA) were used.TfR functionalized nanoparticles, already were commercial particles functionalized available (cat. GCT-50, Creative Diagnostics), while for EpCam functionalized particles a conjugation kit was used (cat. GCK-50, Creative Diagnostics). The conjugation reaction was performed according to the manufacturer’s protocol.
This reviewer was unable to locate any of the catalogue numbers mentioned. No further characterization is included.
All nanoparticles were from CD Bioparticles, see section 2.4 first paragraph. We made a mistake indicating Creative Diagnostics instead of CD Bioparticles. In the CD Bioparticles web, using the catalogue number you can find all particles and kits used.
Results
In the results sections, the organization reflects the techniques/assay used and not the biological results and its relation with the action of the nanoparticles. This reviewer suggests that this should be improved.
As indicated above the entire manuscript has been modified, in the results section the organization follows the same organization but now we believe that the biological results are better explained.
In the section, the characterization of the aunps presented, regarding physical-chemical parameters like size (for instance by TEM), zeta potential and hydrodynamic size should be presented . This is particularly important when comparing the naked with functionalized aunps.
Additionally, there is no information of the ability of the functionalized Aunps to interact with their target receptor, nor about the predicted mechanisms of internalization endocytosis?).
The zeta potential has been included in the text and a new figure (4) has been included. See section 3.3.
“In Figure 4 are shown the ζ-potentials of non-functionalized (NF-GNP) and functionalized GNP (TfGNP and AntiEpCamGNP), measured in water and in culture media. As can be seen, in water NF-GNP showed the most electronegative ζ-potential, a difference that disappeared in the culture medium where the ζ-potentials were less electronegative than in water.“
About the internalization of different GNPs, in the discussion section we have included different factors influencing this item. We think that these paragraphs give a better understanding of this issue. It is well known that Tf and its receptor are internalized by clathrin coated vesicles and recicles to the plasma membrane from endosomes. In the case of EpCam, it has been also described that can be endocytosed and recycled to the plasma membrane in murine teratoma cell lines embryonic stem cells during differentiation and in a human esophageal squamous cell carcinoma (https://doi.org/10.1016/j.isci.2021.103179). However, because our study was not focused on studying the mechanisms of internalization our contribution would be only speculative.
3.1
What was the rationale behind the cytogenetic characterization of the 2 cell lines? These results lack proper integration in the overall context of the study. If this is not provided, then this subsection should be removed.
There are two main reasons for the cytogenetic characterization, first to assess a possible chromosome structural instability that could influence the foci analysis. In addition, assessing the cell variability in the number of chromosomes serves us to consider the analysis of foci in a large number of cells to avoid any bias. Cells with less or more DNA will tend to have less or more foci.
Figure 1 – no scale bar, image with benefit from arrows pointing to the acentric/dicentric chromosomes mentioned in the text.
Scale bars have been included
3.2 Aunps internalization
No naked Aunps were detected, authors should elaborate better on the reasons for this results – if it is a technical question of microscope resolution why was it not the case for the functionalized aunps?
According to the reviewer comments several changes have been done in the manuscript. Now it is specified that non functionalized GNP are citrate stabilized.
This reviewer finds that alternative methods such as ICP-MS should be performed to quantitatively evaluate the internalization of both types of aunps.
Unfortunately, this is the only point addressed by the reviewer that we cannot respond to.
Figure 3 – several corrections are required. No scale bar is presented, and the in the legend is stated “Nuclei are marked red, cytoplasmic membranes are marked blue, and nanoparticles are visualized as green dots”. This is a mistake, as nuclei are blue and cytoplasmic membranes are red.
Figure 3 has been changed entirely (now figures 5 and 6) and the mistakes detected by the reviewer have been corrected.
Additionally, several points need to be addressed:
- Is the cytoplasmic membrane staining the result of the staining for the EpCam and Trf receptors?
Yes, both receptors, as mentioned above, can be internalized by the cells.
- Why are the functionalized aunps visualized as green dots? Where the aunps modified with a fluorescent moiety?
GNPs images were obtained by reflection and a green color was assigned automatically to be distinguished from the blue (nuclei) and red (membranes). This is now explained in the text and in figures 5 and 6 captions.
- In panel 1A, what is the source of the diffuse green fluorescence in the cytoplasm?
As mentioned, the figure has changed. The diffuse green fluorescence was done to light reflection during image capture.
- In panels 2B and more so in panel 2C what are the black structures detected intracellularly? Is it some form of aunps aggregates?
Yes they were aggregates but now this figure has been substituted by figures 5 and 6.
- For Caco2 the images are not of enough quality for publication. These images should be replaced.
As mentioned, images have been changed.
3.3 H2Ax foci
- This is not suitable for an entry in the results section.
The entry is now ᵞ-H2AX foci induction and kinetics.
- A representative microscopy image would have been useful.
Microscopy images of nuclei containing ᵞ-H2AX are now included (Figure 7).
- The graphs and the table present the same information. There is no need to include both. However, looking closely, there is a discrepancy between Figure 4B and Table 1 - Aunps at 2h. This needs to be corrected.
The table has been eliminated and foci and MTT figures have been modified to be clearer.
- What do the authors consider the basal level? Is this the cell line before irradiation?
Yes, this is now indicated in figure 7 caption.
- A relevant control is missing – there is no info about the number of foci in non-irradiated cells at the same time points.
In studies of foci kinetics after irradiation, it is not usual to analyze unirradiated controls at the same post irradiation times of irradiated samples. This is because the basal foci frequencies did not change within 24 hours in culture unless an external clastogenic agent acted.
- Why the large difference in the basal level of the 3 conditions for SKBR3?
The basal frequencies in SKBR3 cells were not significantly different.
3.4 MTT Cell viability assay
The entry needs to be renamed.
The authors need to clarify the difference between cell viability and survival. In most of the reported literature about the radiobiological effects of radiation, the survival is assessed by the clonogenicity assay, in which the ability of the cells to survive and expand after irradiation is determined. The MTT assay measures the metabolically active cells, assumed to be viable. It does not detect the survival ability, as cells viable might not be able to divide and proliferate.
We agree with the reviewer and the text and figure have been entirely modified. Now section 3.5.
“Cell viability was measured by MTT analysis 24 h after irradiation (Figure 10). For both cell lines, without irradiation there were no significant differences in viability between cells grown without GNPs and cells grown with NF-GNP or functionalized GNPs. However, for both cell lines the highest viability was observed for cells grown without GNPs. As expected, after irradiation the viability decreased significantly in all cases (p<0.05). Moreover, for both cell lines the decrease in viability was more pronounced for cells irradiated in the presence of functionalized GNPs but was only significant for Caco2 cells (p<0.05). After irradiation, SKBR3 cells always showed significantly lower viabilities than Caco2 cells (p<0.05). The sensitization enhancement ratio for viability was also higher than one for functionalized particles (Table 1).”
As mentioned, the methods, it is not clear the protocol followed, 24 h aunps, irradiation, plating and then 24 h later MTT?
As indicated above, we have clarified this protocol in section 2.7.
Figure 5 – same scale??
We have modified the text to clarify the protocol.
Discussion
The Initial sentences discussing Aunps in general would be better integrated in the introduction
Discussion of advantages of methods selected – H2Ax and MTT – can be more sucint.
Line 360 – authors speculate about DNA damage induced by Aunps. Needs further discussion with reference to literature supporting this idea.
Line 374 – Nor clear the rationale
We have extensively modified the discussion section according to the reviewer's suggestion and added more references to help to clarify our results.
Reviewer 2 Report
In their manuscript entitled "Differential radiosensitizing effect of 50 nm gold nanoparticles in two cancer cell lines", authors investigated the radiosensitization effect of gold nano-object (functionnalized or not). Please find below my comments :
* Introduction, line 50 : 70% of indirect effect was only reported after X-ray irradiation. Specify this point in the mansucript
We have modified the text to clarify it.
“A radioprotective agent reduces the damage produced in healthy cells by limiting the action on DNA of reactive radicals like reactive oxygen species (ROS), mainly generated by the interaction of ionizing radiations and water [3]. A pioneer study showed that the fraction of the DNA damage that can be protected from radiation by the use of radioprotectors, accounts for about 70% [4].”
* Introduction, line 51 : "... radiosensitizing agents act enhancing the effect of ROS generated by radiation". This is a very simplistic view because there are many other modes of action by which molecules exert a radiosensitizing effect.
We agree with the referee that enhancing the effect of ROS is not the unique way of radiosensitizing. And the use of some chemotherapeutics also enhances radiation effectiveness. We have reformulated the sentence.
“On the contrary, one of the main mechanisms of action of radiosensitizing agents is to enhance the level of radiation induced ROS to cause damage to the DNA of tumoral cells. As radiosensitizing agents, various agents such as hyperbaric oxygen [7], nicotinamide and carbogen [8] are widely described and applied.”
Moreover, we have also modified the discussion to consider other items affecting the radiosensitizing effect of GNPs like size, interaction with oxidative stress control mechanisms or intrinsic cell lines characteristics.
* Introduction, line 65. The reference 12 is from 6 years ago. A more state-of-the-art review will provide more relevant information in light of the knowledge evolution in the field in recent years. I can recommend this one (doi.org/10.1088/1361-6560/ab9159) which is a collaborative work of a series of experts.
In agreement to the referee’s comment we have included new references.
“The effect of nanoparticles as radiosensitizers has been extensively reviewed [12; 13; 14; 15; 16], and their importance as therapeutic agent has been steadily rising with time.”
* Introduction, line 66 : "The most efficient particles seem to be high Z ..." This is not true. As you discuss later, it is very complicated to compare studies due to the large number of experimental parameters involved. Linking radiosensitizing effectiveness to the Z of the metal used implies that the mechanism of action is physical in nature. Although this has been claimed for many years, it turns out that a significant biological contribution is present. Efficacy will therefore depend on the Z of the nanoparticle but also on its capacity to inhibit a series of given biological targets (doi.org/10.3390/cancers12082021). The paragraph needs to be completely rewritten to integrate these mechanistic considerations (physical and biological) in order to give a more up-to-date view of the field. Moreover, your results should be discussed on this biological basis (expression level of the target enzymes (thioredoxin reductase, ...) in your cellular models).
The paragraph has been rewritten to integrate physical and biological mechanisms
“It has been described that high atomic number (Z) materials absorb more energy per unit mass than water when irradiated with X-rays [16]. In the case of gold (Z=79), it can be 100 times more effective at absorbing photon energy than water. This local absorption triggers the emission of low energy electrons from gold with a potential increase of DNA damage. The increase in the effect of a dose when it is delivered in the presence of gold nanoparticles (GNPs) is called “sensitization enhancement ratio” (SER). So, if GNPs can accumulate in specific tissues, it opens the door to a differential enhancement in tumoral tissues [13], allowing for lower radiation doses to have the same effect as higher ones [17, 18]. However, it has been observed that GNPs induce higher SER than could be expected according only to their physical conditions [16; 19; 20]. This means that several factors influence the radiosensitization effect of GNPs. In relation to energy deposition, low-energy X-rays produce a higher enhancement than high-energy X-rays [21, 22] and, the higher the diameter of GNPs, the lower the deposition of energy and the emission of low-energy electrons [23]. On the other hand, the radiosensitizing effect of GNPs is, partially, triggered by an increase in the formation of reactive oxygen species (ROS) when compared with cells irradiated in the absence of GNPs. ROS can react with DNA, inducing double strand breaks and affecting the cell viability. Moreover, without irradiation GNPs also increase the oxidative stress of the cells by interfering in the activity of some antioxidant enzymes [24; 25; 26; 27]. Because ROS have a limited lifespan, it seems reasonable that more radiosensitizing effect is expected if GNP are inside the cell. A higher cell uptake of 50nm diameter GNPs when compared to 14nm or 74nm GNPs has been described [22].”
* Introduction line 71. Reference 16 has been updated in the aforementionned manuscript.
The suggested references have been included.
* Mat&Met line 147 : Please correct 137 Cs in 137 Cs (superscript); Gy.min-1 (superscript) and specify the energy peak of gamma ray emitted by cesium.
Cesium-137 has been corrected and the energy peak has been included.
“Cell cultures were irradiated with gamma-rays 24 h after incubation in the absence or in the presence of nanoparticles (NF-GNP for both cell lines and TfGNP for SKBR3 cells and anti-EpCamGNP for Caco2 cells). Irradiations were done at 3 Gy with a 137Cs irradiator (IBL437C, CIS Biointernational, GIF Yvette, France) located at the Technical Unit for Radiation Protection of the Universitat Autònoma de Barcelona. Dose rate was 5.02 Gy·min−1 and the energy peak was 662 keV.”
* Section 3.2.; line 203. You are limited by the precision of your optical microscope. So what you see is just the gold aggregates. Drawing conclusions on internalization based on these images is therefore a nonsense because you are only focusing on these aggregates while omitting the individualized nanoparticles. Moreover, you do you quantify.
The aim of our confocal observations was not to quantify, but to have a qualitative observation of the different treatments. We have included a text in the material and methods section.
“Cell cultures of both cell lines were treated with 50 nm NF-GNP. Moreover, CaCo2 and SKBR3 cell cultures were treated with AntiEpCamGNP and TfGNP respectively. In all cases, functionalized and non-functionalized GNPs, diluted in 1X PBS, were added to the cell cultures at a final concentration of 7x105 nanoparticles·mL-1. Nanoparticle internalization was evaluated 24 h after treatment of the cell cultures with a broadband confocal microscope (Leica TCS SP5, Leica Microsystems, Wetzlar, Germany) located at the Microscopy Service of the Universitat Autònoma de Barcelona. CellMask™ Deep Red (cat. C10046, Thermofisher, Waltham, Massachusetts, USA) was used to stain cell membranes and Hoechst 33342 (cat. H3570, Thermofisher) to stain cell nuclei. Staining was applied 5 minutes prior to observation. Due to their reflective nature, no additional staining was required to visualize GNPs.”
And we have modified the text in the results section .
“A qualitative assessment of GNPs internalization through confocal microscopy imaging allowed us to detect certain differences between the cell lines (Figures 5 and 6). As can be seen, for both cell lines the internalization of functionalized GNPs (TfGNP for SKBR3 and AntiEpCamGNP for Caco2 cells) is more visible than that for NF-GNPs. Moreover, it seems that SKBR3 cells showed more internalized functionalized GNPs than Caco2 cells (Figure 6). Finally, GNPs aggregation was clearly visible for TfGNP in SKBR3 cells. Despite the different degrees of internalization between cell lines, both showed a common trait, that is the inability of GNPs to enter the nucleus, thus remaining in the cytoplasm.”
* Section 3.3. Please correct the foci value based on significant digits. Example : 17.02+/- 1.05 ==> 17 +/- 1
We have unified and now all means and SE only show one decimal digit.
* Figure 4 : This figure needs to be completely rewritten. Using a box plot will allow the reader to have more information (mean, median, standard deviation) on your data than what you currently report (a mean without standard deviation). The kinetics must be analyzed on a mathematical basis with adapted models to conclude that there is no difference in post-irradiation damage induction. For this, I refer you to the section "Modeling DSB formation and repair" of the review doi.org/10.1093/narcan/zcab046 .
Figures 8 and 9 have been modified and now foci results are shown using boxplot figures.
Finally, the delivered dose is not specified in the legend. If 3 Gy were delivered as suggested in the material and method, it is difficult to conclude anything from these results. Indeed, above 1 Gy, there is a loss of correspondence between the number of DSBs and the number of foci driven by the clustering phenomenon (cf. section "dependence of dose" in the aforementionned review).
We agree that increasing the dose the clustering phenomenon may underestimate the foci frequency. However, in the present study γ-H2AX-foci analyses an automated scanning fluorescence microscope system was used and processed using the MetaCyte software. Foci images were captured using a 63_PlanApo objective. The foci signals in the selected nuclei were captured using the SpOr filter (red channel). All the SpOr signals were acquired as a z-stack with a total of 10 focal planes and a z-step size of 0.35 μm between planes. These focal planes reduce the clustering phenomenon.
“γH2AX foci analyses were done using an automated scanning fluorescence microscope system (Metafer 4, Meta Systems, Altlussheim, Germany) and processed using the MetaCyte software (version 3.10.2) coupled to a motorized z-stage Zeiss Axio Imager.Z2 microscope (MetaSystems) The images were captured using a 63_PlanApo objective. The foci signals in the selected nuclei were captured using the SpOr filter (red channel). All the SpOr signals were acquired as a z-stack with a total of 10 focal planes and a z-step size of 0.35 μm between planes. A unique classifier was used to count foci in about 2000 nuclei for each experimental condition.”
* Figure 5 : Statistic is missing. As claimed in your manuscript, MTT is a bioassay that enable to analyze cell survival or cell proliferation. You analyzed survival rates at 1 time point post-IR and one dose. So, how can you say that the effect you reported is a decrease in cell survival and not in cell proliferation ?
We agree with the reviewer that our text was confusing. It has been corrected and only viability rates are shown.
Reviewer 3 Report
The current manuscript focused on the application of non-functionalized and functionalized AuNPs as radio-sensitizers. Two cell lines which differ in their radio-sensitivity were used to confirm the thesis that no universal radio-sensitizing effect for AuNPs on all cell lines could be defined. Despite the interesting thesis, there are some important questions that remained unanswered and are some concerns regarding the execution of this study.
● There are no “naked” AuNPs. According to the manufacturer the “non-functionalized” AuNPs are either stabilized with citric or tannic acid.
In agreement to the reviewer’s comment we have clarified in the text that non-functionalized AuNPs were citric acid stabilized. Material and methods section 2.4.
“In the present study three types of GNPs were used. Citric acid stabilized 50 nm GNPs (cat. BG-50, CD Bioparticles, Shirley, NY, USA) from now on non-functionalized gold nanoparticles (NF-GNP), commercially Tf functionalized gold nanoparticles (TfGNP) (cat. GCT-50, CD Bioparticles) and anti-EpCam antibody functionalized gold particles (anti-EpCamGNP) functionalized with a conjugation kit (cat. GCK-50, CD Bioparticles) according to the manufacturer’s protocol.”
● For the functionalized AuNPs, two different NPs are described in the experimental section: TfR-functionlized AuNPs (directly ordered) and EpCAM AuNPS (functionalized by a kit). Which one is the one mentioned in the manuscript as functionalized AuNP? When both are used: Why the results are not separately displayed?
We have clarified in the second paragraph of section 2.4 that both cell lines were treated with functionalized AuNps. For SKBR3 the treatment was with Tf-functionalized AuNPs (TfGNP in the revised manuscript), and for CaCo2 cells with anti-Epcam AuNPs (anti-EpCamGNP). See above indicated sentence.
● There is no further characterization of the AuNPs like their stability in water, agglomeration in medium. At least some DLS measurements should be shown.
We have included in the revised version, the ζ-potential values for all GNPs both when diluted in water and in culture media. 2.4 section and figure 4.
“To measure the ζ-potential, functionalized and non-functionalized GNPs were separately resuspended in water or in EMEM (anti-EpCamGNP) and McCoy’s 5A (TfGNP) culture mediums and sonicated for 5 min (Fisherbrand FB15047, Fisher Scientific, Germany) to achieve a monodispersed sample. The ζ-potential was then measured with a Zetasizer Nano ZS (Malvern Instruments, Malvern, UK).”
● The concentration of the NPs in the experiments is stated as 7x105 According to the manufacturer concentration of the 50 nm AuNPs (0.05 mg/mL) or TfR-AuNPs (130 μg/mL). How did you determine the NP concentration? Why didn´t you use the concentration in μg/mL?
We agree with the reviewer that probably is more accurate to work with NP concentration. However, because we used the same GNPs diameter (50 nm) that Chithrani et al 2010, we decided to use also a number of GNPs scaled to our experimental conditions.
● The cellular uptake study via fluorescence microscopy is rather incomplete:
● a) why do the functionalized AuNP suddenly fluorescent green in the microscope images?
This issue has been clarified in the revised manuscript (last sentence of 2.4 section and figures 5 and 6 caption). NPs weren’t fluorescent, but to a better visualization to their reflected light, an artificially green color was assigned.
● b) Due to their functionalization the functionalized AuNPs should be internalized via different receptors. How do the cell lines differ in these receptors and is that not the reason for their different internalization rate? c) You stated that the resolution is too low for imaging the AuNPs. Why not using TEM? Why not determine the Au concentration per cell via ICP-AAS or something similar?
The aim of our work was to study if GNPs have a sensitization effect when cells are irradiated in their presence and not to describe the ways of GNPs internalization. The use of specifically targeted and non-specifically targeted GNP allows to know if targeting could be a useful way for radiosensitization. Moreover, It is well known that Tf and its receptor are internalized by clathrin coated vesicles and recycles to the plasma membrane from endosomes. In the case of EpCam, it has been also described that can be endocytosed and recycled to the plasma membrane in murine teratoma cell lines, embryonic stem cells during differentiation and in a human esophageal squamous cell carcinoma (https://doi.org/10.1016/j.isci.2021.103179). However, because our study was not focused on studying the mechanisms of internalization our contribution would be only speculative.
● The different radio-sensitizing effect on the cell lines is due to a difference in internalization of the functionalized AuNPs. What is the impact of the different radio-sensitivity of the cell lines?
We have better clarified these two aspects in results and discussion sections.
According to the reviewer's suggestion, we have extensively modified the discussion. New paragraphs have been added, where physical GNPs properties, but also biological considerations on GNPs biofunctionalization or intrinsic cell lines uptake capacities or radiosensitivity were discussed.
Reviewer 4 Report
Cancers (ISSN2072-6694)
Title: Differential radiosensitizing effect of 50 nm gold nanoparticles in two cancer cell lines
The manuscript describes the radiosensitizing effect of 50 nm spherical gold nanoparticles, both naked and functionalized, in two different cell lines, Caco2 (colon adenocarcinoma) and SKBR-3 (breast adenocarcinoma). Experiments are well done and convincing. However there is an important problem with this paper. In lines 206-208 the authors indicate the following trend in feasibility tests: no AuNP > naked AuNP > functionalized AuNP.
We suppose that the reviewer refers to lines 286-288 of the original article where regarding viability test it was stated “a significant difference (p=0.002) was found between irradiated cells without nanoparticles and irradiated cells treated with functionalized nanoparticles.” This is exactly what it was expected. The viability after irradiation in the presence of functionalized GNP was lower when compared with irradiations in the presence of non-functionalized GNP or in absence of GNPs. These results indicate a higher radiosensitizing effect of functionalized GNPs. However to clarify this item, a new table has been added with sensitization enhancement ratio values.
On the other hand authors assume that: “while 50 nm AuNP induce a radiosensitizing effect, it is highly difficult to describe this effect as universal since a high heterogeneity was found between cell lines”. In this sense, I would like to comment that if the conclusions are not at all definitive, they should clearly indicate the novel contribution of their work.
We agree with the reviewer that the original manuscript was somehow confusing. We have now clearly stated the radiosensitizing effect for foci and viability of functionalized GNPs in section 3.4, where you can find the table 1 and the following sentences in section 3.4.
“At any postirradiation time, the frequencies of foci in cells irradiated in the presence of AntiEpCamGNP were significantly higher than for the other two treatments (p<0.001).”
“At all postirradiation times the frequency of foci was significantly higher for cells irradiated in the presence of TfGNP (p<0.01).”
On the other hand, the fact that the viability of the functionalized nanoparticles differs from the naked ones, let alone their absence, is not a novel contribution, since the second principle of thermodynamics itself says so: an improvement requires an effort and a functionalization of a nanoparticle is an effort seen from the point of view of work.
The improvement of cancer treatments to make them more efficient, needs an important effort. Targeting GNPs, to selectively kill tumour cells is without doubt a worthwhile effort. The same happens with new therapeutic strategies like immunotherapy where in some cases targeting specific antibodies increase the survival rates.
There is another substantial problem. The authors use commercial nanoparticles but do not refer at any time to a previous characterization. It is understood that they are monodisperse samples and that they are perfectly characterized, although a previous study of the sample is advisable for a job of this magnitude. This point must be described.
According to the reviewer comment, we have characterized the zeta potential of non-functionalized GNPs and of transferrin functionalized GNPs and Anti EpCam antibody functionalized GNPs. The ζ-potential has been measured in water and in cell culture medium because serum proteins can modify it. These values can be seen now in figure 4 and discussed in the following paragraph included in the discussion section:
“It is well known that the surface charges of nanoparticles influence their internalization by the cells, being highly charged nanoparticles (positively or negatively) more uptaked by cells than neutral ones. It has also been described that functionalization of polystyrene macroparticles to make them positively charged increases significantly their internalization by HeLa cells [59]. We observed that the ζ-potential of all GNPs (functionalized and non-functionalized) become less electronegative when the ζ-potential was measured in culture medium unless in water. Moreover, the differences observed in water disappeared in cell culture medium. This could be due to a protein corona formation, with serum proteins, when particles are in culture medium. This modification has been suggested to be the responsible of the decrease in particles uptake by the cells [60, 61], but it has also been described that the impact can vary between cell types [62].”
On the other hand, why do they work with 50nm?, It is just the limit (25-50nm) of the value indicated by reference 17.
We choose to work with 50nm GNPs for two reasons. The first is that it was described that 50nm GNP are more uptaked by cells and show a higher sensitizing effect than smaller and larger GNPs (see references 22 and 63). The second is because working with three types of GNPs (the non-functionalized and the two functionalized ones) and two different cell lines is of enough complexity to add more influencing factors.
The authors must make an effort considering the quality and impact of the journal. In these conditions the paper must be revised pointing out the real improvements and contributions to the study and not just describing it as good experimental work.
According to the reviewer's suggestion the entire manuscript has been rewritten. We believe that now our contribution has been improved.
Minor points: there are references that can be replaced by much more current ones. Although some such as 8 and 9 refer to a very specific topic and it is not easy, there are others such as 14 that can be clearly updated.
The reviewer comment has been considered and an exhaustive bibliography revision has been done, incorporating 33 new and more current references.